# Approaches for sRNA Analysis of Human RNA-Seq Data: Comparison, Benchmarking

**DOI:** 10.3390/ijms24044195

**Published:** 2023-02-20

**Authors:** Vitalik Bezuglov, Alexey Stupnikov, Ivan Skakov, Victoria Shtratnikova, J. Richard Pilsner, Alexander Suvorov, Oleg Sergeyev

**Affiliations:** 1Belozersky Institute of Physico-Chemical Biology, Lomonosov Moscow State University, 119992 Moscow, Russia; 2Faculty of Bioengineering and Bioinformatics, Lomonosov Moscow State University, 119992 Moscow, Russia; 3Department of Biomedical Physics, Moscow Institute of Physics and Technology, 141701 Moscow, Russia; 4National Medical Research Center for Endocrinology, 115478 Moscow, Russia; 5Department of Obstetrics and Gynecology, Wayne State University School of Medicine, Detroit, MI 48201, USA; 6Department of Environmental Health Sciences, University of Massachusetts, Amherst, MA 01003, USA

**Keywords:** sRNA analysis, small RNA, microRNA, piRNA, tRNA-derived small RNA, RNA-seq, small RNA fragments, benchmarking, differential expression analysis

## Abstract

Expression analysis of small noncoding RNA (sRNA), including microRNA, piwi-interacting RNA, small rRNA-derived RNA, and tRNA-derived small RNA, is a novel and quickly developing field. Despite a range of proposed approaches, selecting and adapting a particular pipeline for transcriptomic analysis of sRNA remains a challenge. This paper focuses on the identification of the optimal pipeline configurations for each step of human sRNA analysis, including reads trimming, filtering, mapping, transcript abundance quantification and differential expression analysis. Based on our study, we suggest the following parameters for the analysis of human sRNA in relation to categorical analyses with two groups of biosamples: (1) trimming with the lower length bound = 15 and the upper length bound = Read length − 40% Adapter length; (2) mapping on a reference genome with bowtie aligner with one mismatch allowed (-v 1 parameter); (3) filtering by mean threshold > 5; (4) analyzing differential expression with DESeq2 with adjusted *p*-value < 0.05 or limma with *p*-value < 0.05 if there is very little signal and few transcripts.

## 1. Introduction

Small non-coding RNA (sRNA) of less than 200 nucleotides in length are important regulatory molecules in the control of gene expression at both the transcriptional and the post-transcriptional level [1,2,3]. Research on sRNAs has accelerated over the past two decades and sRNAs have been utilized as markers of human diseases such as neurological conditions [4], cancer [5] and infertility [6,7,8], and in the identification of molecular biomarkers associating environmental exposure with health/disease outcomes [9,10]. The identification of sRNA in germ cells is of particular interest as they represent an additional source of parental hereditary information beyond DNA sequences and may have a potential role in programming offspring health [11,12]. Types of small RNA include, among others, microRNA (miRNA), piwi-interacting RNA (piRNA), small rRNA-derived RNA (rsRNA) and tRNA-derived small RNA (tsRNA) [2]. Next generation sequencing (NGS) has become the principal approach for the global profiling of sRNA due to the steady decrease in sequencing costs, and their wider coverage and higher sensitivity compared to microarrays. However, the bioinformatic analysis of NGS data for sRNA is prone to many challenges. For example, the calculation of sRNA expression values from NGS reads may not reflect their absolute expression levels accurately [13,14]. In this study, we attempt to identify the optimal pipeline configurations for each step of the sRNA analysis of human data, including read trimming, filtering, mapping, transcript abundance quantification, and differential expression (DE) analysis.

### 1.1. sRNA Expression Methods

All major pipelines for sRNA expression analysis may be classified into three groups. First, tools that only allow for a particular stage of expression analysis (namely, alignment). Second, the ones that allow for the more expanded analysis of a certain type of sRNA (piRNA or miRNA, for instance). The tools of the third group provide a means for the expanded analysis of two or more types of sRNA. The characteristics of some existing sRNA pipelines are presented in Table 1.

The two tools that are specialized in the mapping of small RNAs only are similarly named: srnaMapper [15] and sRNAmapper [16]. Both have regular alignment options. srnaMapper has the additional ability to align reads without precise complementarity in the last few bases. This, however, may result in a reduction in mapping specificity and a significant increase in the required computational resources.

Several tools in the second group were designed specifically for piRNA annotation of NGS-samples, namely, piPipes [17], PILFER [18], piClust [19], proTRAC [20]. Pipelines focused on miRNA detection include miRanalyzer [21] and sRNA workbench [22]. tsRFun [23] is focused on tsRNA analysis. As each of these tools focuses on a single biotype of sRNA, we do not consider them further in this study.

A number of tools are available for the analysis of several different types of sRNAs. These are sRNAtoolbox [24], sRNAnalyzer [25] and SPORTS [26], which have command-line-based interfaces (CLI). Other tools have a graphic-based user interface (GUI): sRNAPipe [27] (Galaxy-based pipeline) and iSmaRT [28]. Oasis 2 [29] offers only an online version, which was not fully functional as of October 2022.

A comparison of the most popular pipelines is presented in Figure 1.

The sRNAnalyzer [25] is a command-line interface pipeline with a text-based configuration file for the identification of miRNAs, piRNAs, small nucleolar RNA (snoRNA), and long non-coding RNA (lncRNA). The pipeline allows for the processing of reads (upon adaptor removal), quality filtering, read mapping and counting. The preprocessing steps include removing adaptors and filtering by the minimum length of the read. For the analyzed sequences, sRNAnalyzer uses a ‘map and remove’ approach with a progressive alignment strategy to sequentially map the reads against various databases (only the reads that are unmapped to the current database will proceed to mapping to transcript sequences in the next database). Databases for the following species are available for use with the sRNAnalyzer: human, mouse, rat, horse, macaque and plant. Moreover, they can be modified for samples of other species. sRNAnalyzer uses FASTQ files for input, while the output provides files with gene counts.

SPORTS1.0 [26] is a command-line-based pipeline for the identification and quantification of miRNAs, piRNAs, tsRNA and rsRNA. SPORTS1.0 can be used with a wide range of species with an available reference genome. It is also possible to substitute the default small RNA databases with custom databases provided by the user [30]. The pre-processing steps include adapter removal and filtering sequences by size and quality. The output is provided as gene counts and various visualization figures.

iSRAP [31] is another tool with a command line interface, focused on the annotation of sRNAs (miRNAs, piRNAs and snoRNAs). A configuration file is needed to define the options and optimize sRNA profiling in different datasets. This pipeline can be executed by starting with either FASTQ or BAM alignment files as input. Output results are reported as PDF files and HTML documents, and graphical elements are used to illustrate the results.

One of the most popular instruments for small RNA annotation is a web-interface-based sRNAtoolbox [24]. This includes several tools: sRNAbench for the expression profiling of small RNAs and prediction of novel miRNAs from deep sequencing data; sRNAde for the DE analysis; sRNAblast for a blast analysis of deep sequencing reads against a local database, and others. Adapter trimming, read quality and size filtering are available and optional in sRNAbench.

The above-mentioned pipelines and others were recently reviewed in detail elsewhere [32].

### 1.2. Alignment-Based Tools

Several pipelines have been developed for RNA-seq data analysis [33,34,35]. Their recruitment in sRNA expression can potentially help to overcome the ’map and remove’ approach problems, and avoid the non-independent processing of different sRNA types.

The standard workflow for RNA-seq analysis includes several steps. First, preprocessing involves reads trimming to remove the adaptors or low-quality bases from reads ends, and this may be carried out using specific tools, such as Trimmomatic [36] or cutadapt [37]. The next step is reads alignment, and this can be carried out with a variety of tools, such as hisat2 [38], STAR [39], bowtie [40], bowtie2 [41].

At the next step, the mapped reads are summarized for a particular transcript annotation. Frequently used approaches for this procedure include featureCounts [42] from Rsubread, and HTSeq [43]. The output of this step is a count matrix that represents the expression values for all considered samples and genes or transcripts. Some approaches may combine alignment and quantification procedures. These include RSEM [44], which performs transcriptome alignment and produces expected counts for transcripts, and probabilistic pseudoaligners, such as Kallisto [45] and Salmon [46], which assign reads to transcripts based on their k-mer spectra pattern and provide estimated counts.

Finally, the obtained counts are normalized and, after filtering, inferred with the statistical model for DE. Commonly used approaches for this step are DESeq2 [47], edgeR [48] and limma [49], or various Bayesian models [50,51]. As a result, the list of DE genes (i.e., gene signature) or transcripts is retrieved.

However, adapting the described genome-alignment-based expression analysis workflow poses several challenges. First, due to the very short length of sRNA transcripts, the influence that the samples and library preparation-related parameters (sequencing kit or adapters choice) have on the result of preprocessing and trimming procedures is higher compared to that of mRNA reads preprocessing. Second, the small length of reads makes the alignment procedure more challenging [52]. Therefore, the performance of various aligning approaches needs to be evaluated considering sRNA data applicability. Third, due to the significantly lower expression signal in sRNA data, the filtering of low-expressed transcripts, aiming to reduce noise, is important [53]. Finally, the choice of DE model was shown to have a significant impact on the results [54,55,56,57]. Hence, a quantitative estimation of the resulting expression signature needs to be conducted.

Thus, the objectives of the current study are as follows. First, to explore the optimal parameters for sRNA-data-specific preprocessing steps (such as the choice of sequencing kit or adapters and trimming threshold). Second, to assess the alingment performance and summarize the sRNA data procedures. Third, to evaluate filtering procedures for lowly expressed transcripts. Finally, to estimate the resulting expression’s signature quality after DE inference.

## 2. Results

We used small RNA sequencing data (SRA archives) from 7 published human studies: dataset from Wong et al. article [58] (hereinafter referred to as “Wong”); dataset from Huang et al. article [59] (hereinafter referred as “Huang”); dataset from Kanth et al. article [60] (hereinafter referred to as “Delker”); dataset from Morgan et al. article [61] (hereinafter referred to as “Morgan”); dataset from Hua et al. article [62] (hereinafter referred to as "Hua"); dataset from Donkin et al. article [63] (hereinafter referred to as “Donkin”); and dataset from Ingerslev et al. article [64] (hereinafter referred to as “Ingerslev”).

### 2.1. Input Data

The nature of human biosamples and differences in their processing pipelines, from library preparation to published fastq-files, result in a variation between datasets [58,59,60,61,62,63,64]. We used metrics (such as biological object, lab kit, read and adapter length, reads quality) for consideration at the first stage of analyses in order to choose suitable pipelines, as presented in tables in Section 4.

### 2.2. Trimming

Based on the distribution and peaks in read length for various datasets, we can observe a high variability between datasets and similar distribution and peaks across samples of the same dataset (for most datasets) (Appendix A). However, the lower variability of read length across samples was observed in ”Hua“, ”Huang“ and “Morgan” datasets. “Delker” reads prepared by NEBNext lab kit have peaks at 17, 22, and 32 nt. These peaks may indicate piRNA, miRNA and tsRNA, respectively. The same peaks are observed in the “Huang” dataset, and the “Wong” dataset (prepared by NEXTFlex). One strong peak at 23 nt is observed in the “Hua” dataset and the “Wong” dataset (prepared by Qiaseq). An additional 15 nt reads peak occurred in “Delker” dataset (prepared by TruSeq) and “Wong” dataset (prepared by CleanTag).

The results for various trimming strategies are presented in Figure 2.

The strategy of removing a fixed number of bases was inconsistent and not conserved across datasets and kit types; thus, it was not feasible to observe an optimal trimming length value while retaining a sufficient number of reads for all observed data (Appendix A). The strategy of removing a particular adapter fraction based on read length/adapter length ratio exhibited a similar pattern problem, with an even lower number of retained reads in most cases (Figure 2).

The strategy of removing a given fraction of the adapter appeared to be the most conservative across all datasets and kits (Figure 2A). With the removal of 40% of adapter length, a sufficient number of reads were retained after the trimming procedure (Figure 2B) and, for this reason, this approach was chosen as the preprocessing pattern for all the samples in all datasets.

### 2.3. Genome or Transcriptome Aligning

Reads were mapped with genome-alignment-based methods (bowtie, hisat2 and STAR) and the prebuilt transcriptome alignment-based method (RSEM) demonstrated sufficient alignment rates (Appendix A). A comparison of alignment rates and the number of mapped reads is given in Figure 3.

The highest alignment rate values were observed for bowtie, with one mismatch allowed (bowtie -v 1) (up to 97% for “Donkin” dataset and mean 89% for all datasets), with no mismatches allowed (bowtie -v 0) as the second highest fraction of aligned reads (up to 86% for “Donkin” dataset and mean 74% for all). RNA-seq specific aligners, STAR and hisat2, and RSEM demonstrated lower mean values for alignment ratios (54%, 61% and 57% respectively). However, for some datasets, hisat2 and STAR had higher alignment rates (74% for “Hua” dataset for hisat2 and 85% for “Huang” dataset for STAR). No alignment rates for RSEM exceeded 70%.

### 2.4. Assigning

Assignment rates and input reads ratios are shown in Figure 4. In the process of assigning mapped reads to transcripts loci, all genome-alignment-based approaches demonstrated similar read ratio values, and were successfully assigned to transcripts (Figure 4B and Appendix A). Bowtie with 1 mismatch demonstrated better results based on moderately higher alignment rate, although its assignment rate remained close to that of other approaches.

We observed large variations in assignment rates across datasets and across samples of several datasets, specifically for “Delker” and “Wong” data, that may be explained by the different kits used for sample preparation within one dataset (Figure 4B–D). The IQR for “Wong” data for bowtie mapper with one mismatch (bowtie -v 1) ranged from 0.1 to 0.5. “Delker” data showed 0.4–0.7 IQR for the same pipeline (Figure 4B).

RSEM showed low assignment rate values for all datasets. The highest mean assignment rate (assigned/input) was 0.26 for “Morgan“ data. This may be explained by the transcriptome assembly process (as a part of RSEM analysis), which was likely not optimized for short sRNA transcripts (Figure 4B).
Figure 2(**A**) Percent of reads passing trimming for all datasets with various upper-length bounds [58,59,60,61,62,63,64]. All reads were previously trimmed of adapters, and reads less than 15 nt were removed (No boxplot). Upper-bound choosing strategy defined by color: 45nt - grey, Readlength−partofadapterlength(R−X%×Adapterlength) - red, Readlength−fixednumber(R−X) - green, Readlength×(1 − XReadlengthAdapterlength) - light blue. (**B**) Number of reads before (black border) and after (red border) trimming among datasets and adapter length (boxplot fill) for analyzed datasets [58,59,60,61,62,63,64]. (**C**) Mean read length after trimming for analyzed datasets [58,59,60,61,62,63,64].
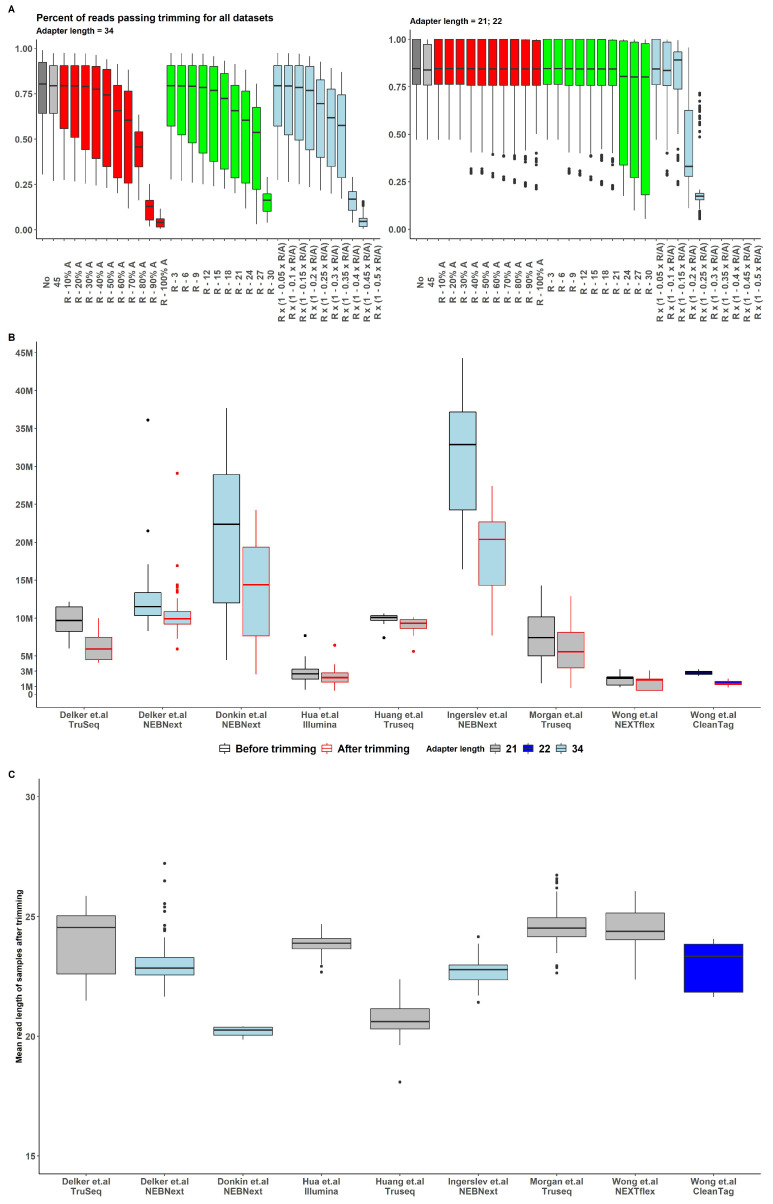

Figure 3(**A**) Percent of reads passing the genome alignment (for bowtie, hist2 and STAR genome aligners) or prebuilt transcriptome (for RSEM) (**B**) Number of reads passing the genome alignment (for bowtie, hist2 and STAR genome aligners) or prebuilt transcriptome (for RSEM) across all datasets [58,59,60,61,62,63,64].
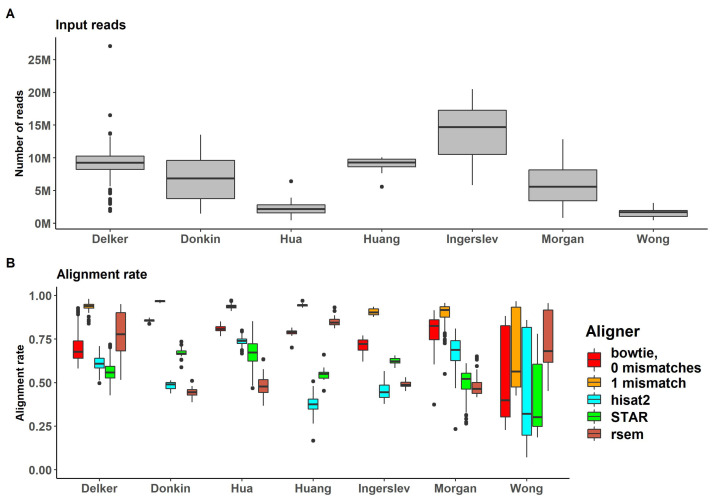



Pseudoaligners kallisto and salmon do not have a specific assignment rate, since reads are probabilistically aligned to transcripts; therefore, the procedure combines the alignment and assignment of reads. As expected, we observed large variations between datasets (Figure 4C,D, Appendix A). The kallisto probabilistic aligner demonstrated various ’aligning ratios’ for different kmer lengths. Kallisto with kmer length = 5 had the highest assignment rate (assigned/input) for all datasets (all samples had an assignment rate of more then 0.5). Kmer length = 7 and 9 led to the lowest assignment rate (Appendix A). These results suggest that the kallisto pipeline with kmer length = 5 may provide many false-positive reads. Salmon pseudoalignment results demonstrated no significant difference for the various kmer lengths (Appendix A).

### 2.5. sRNA Biotypes Distribution

Figure 5A,B demonstrates the distribution of sRNA expression values according to the sRNA biotypes in all pipelines and datasets, and the distribution of the numbers of transcripts expressed by sRNA biotypes in a similar manner, respectively. There is a large variability between datasets and pipelines in both expression values and the number of transcripts. Most datasets, except “Huang” and “Wong”, show a relatively similar sRNA distribution across pipelines. There are also differences between alignment-based, pseudoalignment-based and sRNA-based pipelines within each dataset, especially for “Donkin” and “Ingerslev” data. Due to the limitations of these pipelines, RSEM did not identify miRNA transcripts in all the considered datasets; SPORTS did not identify piRNAs; and sRNAnalyzer did not identify tRNA transcripts. The distribution of the number of expressed transcripts by pipelines is shown in Appendix A. Some pipelines, such as SPORTS, RSEM, and kallisto with kmer lengths 5, 7, and 9, performed worse than others using the methods and criteria outlined in this study.
Figure 4(**A**) Number of reads after trimming procedure across all datasets [58,59,60,61,62,63,64]. Percent of reads processed by (**B**) alignment-based pipelines; by (**C**) kallisto; by (**D**) salmon; by (**E**) small RNA based pipelines.
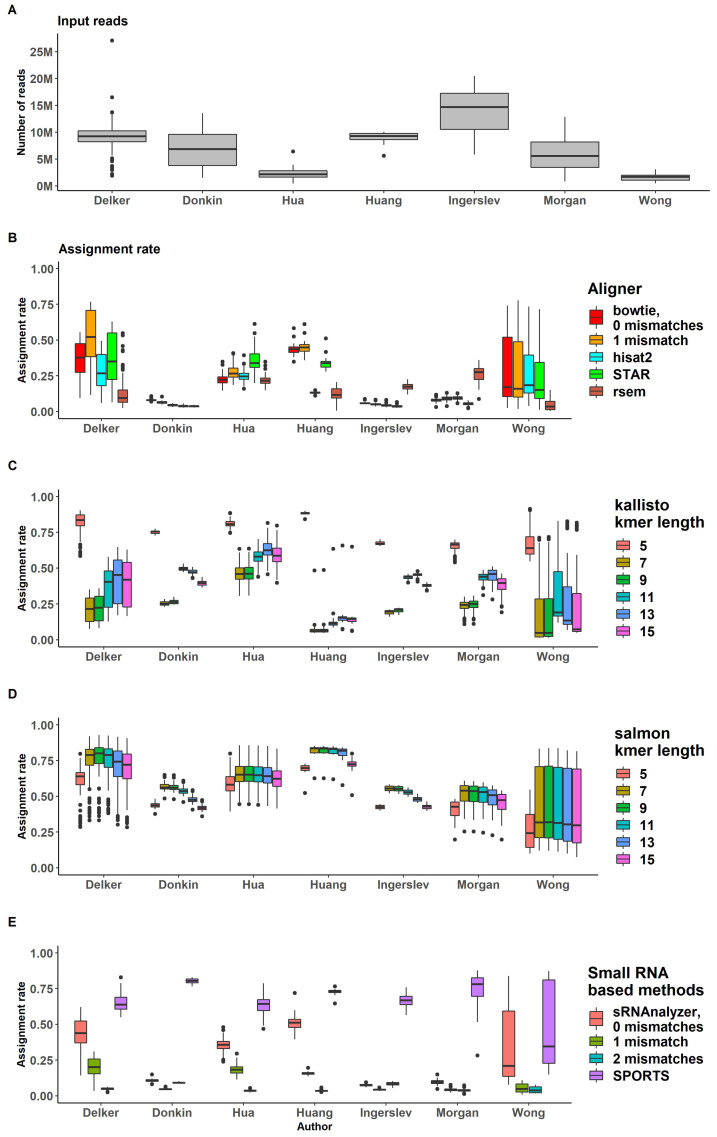


### 2.6. Filtering

Transcript numbers resulting from different filtering strategies (thresholds) are presented in the Appendix A for each dataset and pipeline, respectively. As expected, the min filtering approach resulted in the lowest number of transcripts. The highest number of transcripts was observed with mean filtering, with the threshold 5. Median filtering with the same threshold 5 and mean (counts) > 10 returned a slightly lower number of transcripts.

The mean (counts) > 5 appears to be the optimal filtering approach for expression processing based on the results of transcript numbers and distribution of sRNA biotypes. Using the mean (counts) > 10 and the median (counts) > 5 may be a preferred choice when high numbers of false-positives are found with mean filtering, with the threshold = 5.

### 2.7. Differential Expression

The results of the DE analysis were obtained using in-built package models DESeq2, edgeR and limma, two filtering options (mean (counts) > 5 and median (counts) > 5) and two thresholds of significance, *p*-value < 0.05 and q-value (FDR-adjusted *p*-value) < 0.05. The numbers of DE transcripts with q-value < 0.05 per dataset and pipeline are shown in Figure 6, and Appendix A.

DESeq2 delivered the highest number of significant DE transcripts with the application of multiple testing corrections filtering; the mean of all pipelines was 20.7 and 12.8, for the mean (counts) > 5 and the median (counts) > 5, respectively. EdgeR returned fewer significant transcripts; the mean for all pipelines was 10.0, which was the same for both thresholds. The lowest number of significant transcripts was produced by limma; the mean for all pipelines was 6.4, which was the same for both thresholds (Appendix A).

The DE analysis of the two groups of contrast samples revealed two types of dataset. “Delker”, “Huang”, “Hua” had a fairly large number of significantly DE transcripts, with the mean number for all methods obtained by DESeq2 = 26.8, 44.5 and 68.5, respectively. The other group included “Donkin” and “Ingerslev” datasets, which had a very low number of significant transcripts; the means for all methods were 0.5 and 0.6, respectively. The means obtained by sRNAnalyzer with two allowed mismatches were 5 and 8, respectively.

We suggest that this grouping of datasets is predictable, based on the biological differences between the analyzed samples. We do not anticipate big difference between groups in both the “Donkin” and “Ingerslev” datasets, because both obesity (“Donkin”) and exercises (“Ingerslev”) were not associated with sperm RNA expression by any known strong causative link. All other datasets were related to a distinct tissue type and disease state: “Delker”—different tissues and neoplasm vs. no neoplasm, “Huang”—health and disease, “Hua”—embryo quality that may be directly related to sperm program.

### 2.8. Expression Signature Quality Estimation

We measured the expression signature quality using the Hobotnica approach, which provides separation values (H-scores), ranging from 0 (the worst) to 1 (the best). Filtering and DE analyses are presented for all datasets in Appendix A and Appendix A. Three packages for DE analysis, DESeq2, edgeR and limma demonstrated similar H-scores, ranging from 0.46 to 0.93 among all datasets and pipelines. Two datasets (controls “Delker-C” and “Huang”) demonstrated good separation quality for genome-alignment-based and sRNA-based pipelines (“Delker-C”: mean 0.93 and 0.79, respectively; “Huang”: 0.76 and 0.75, respectively). Other datasets revealed similar and lower H-scores. Overall, genome-alignment-based methods usually produced signatures with higher H-scores (Figure 7). DESeq2 with mean >5 filtering demonstrated a significant difference between pipeline groups. Pipelines based on pseudoalignment provide smaller H-scores for all DE analysis methods (Appendix A). Limma delivered a higher H-score using DE transcripts without FDR *p*-value adjustment (Appendix A).

### 2.9. Transfer RNA Fragments Analysis

For tsRNA expression, read alignment and quantification patterns were similar to those of other sRNA types. Reads assigned to mature tRNA by Rsubread were pseudoaligned by Kallisto to tsRNA sequences. High assignment rates are presented in Appendix A. The pipelines bowtie without mismatches (bowtie -v 0) and hisat2 revealed the highest alignment rates across all estimates (mean = 97%), while STAR demonstrated the lowest assignment rates (mean = 79%). The ratio of assigned tsRNA reads and all input reads is shown in Appendix A. The hisat2 pipeline demonstrated the highest rates across all estimates (mean = 4.6%), while the lowest assignment rates were STAR (mean = 2.7%), and the two variations in bowtie −3.3% of reads (Appendix A).

A very low number of significant tsRNA was found using edgeR and limma across all datasets (up to a mean of 2.7 for all pipelines with mean >5 filtering and adjusted *p*-value < 0.05) and across all pipelines (up to a mean of 1.4 for all datasets with mean >5 filtering and adjusted *p*-value < 0.05) (Appendix A). The number of significant tsRNA using DESeq2 was greater; the mean for all pipelines was 32.0 and 2.5, with mean >5 and the median >5, respectively.
Figure 7Distribution of H-scores across pseudoalignment-based pipelines (kallisto and salmon), sRNA-based pipelines (SPORTS and sRNAnalyzer) and alignment-based pipelines (bowtie, hisat, STAR and rsem) for every dataset [59,60,62,63,64] (**A**) and for all data (**B**).
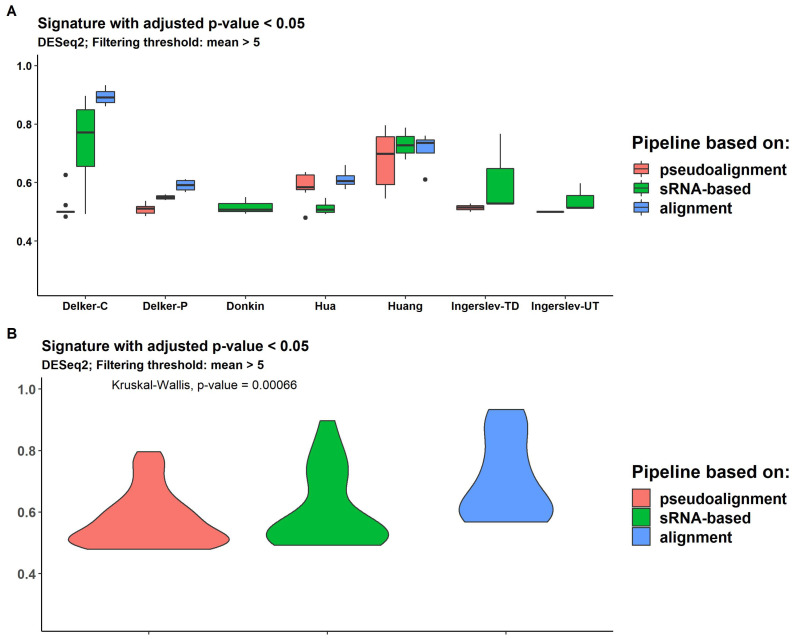



An H-score of higher than 0.7 was only observed for the gene signature obtained by limma with a non-adjusted *p*-value < 0.05 (Appendix A). The highest H-score for an adjusted *p*-value signature was 0.68 (”Delker-C“, filtering by mean (count) > 5, DESeq2). H-scores higher than 0.6 were only obtained by DESeq2. These values were lower than those observed for sRNA transcripts’ expression analysis, which may be explained by the lower tsRNA signal compared to sRNA.

### 2.10. sRNA Tools Performance

SPORTS provided higher assignment rates than other pipelines for almost all datasets (Appendix A). For ”Delker” and “Wong”, the data results were similar to the bowtie aligner rate. The sRNAnalyzer without allowing for mismatches demonstrated a higher assignment rate than when mismatches were allowed (mean = 0.28 versus 0.1 and 0.05 for sRNAnalyzer with 1 and 2 allowed mismatches, respectively), but less than SPORTS (mean = 0.67) (Figure 4). SPORTS and sRNAnalyzer signatures had a slightly lower H-score than alignment-based pipelines across all datasets (Figure 7B). However, for “Donkin” and “Ingerslev”, sRNA-based pipelines managed to provide gene signatures with a higher H-score than pseudoalignment- or alignment-based approaches (Figure 7A and Appendix A).

### 2.11. Overall Performance and Recommendation

Each stage of sRNA analysis, as indicated earlier, can be conducted by employing various methods and different parameter settings for each method. Nevertheless, some parameters sets or tools may produce better results than others when using metrics for each analytical stage of benchmarking. Thus, we suggest the following pipeline, as described in Table 2. The trimming procedure with the flexible upper bound (Read length − 40% Adapter length) demonstrated more stable results when assessed by reads after trimming across datasets. The bowtie aligner with one mismatch (bowtie -v 1) demonstrated the highest alignment rate for sRNAs across all datasets. Filtering using a mean count higher than 5 and applying DESeq2 led to higher H-scores for the obtained gene signatures.

Using main metrics/benchmarks, we presented the results of the application of this pipeline for all datasets in Table 3.

## 3. Discussion

We used seven publicly available human datasets and metrics/benchmarks for each step of sRNA-seq data analysis, from biosampling and library preparation to DE analysis in an effort to produce an optimized sRNA-seq pipeline. Based on our analysis, we suggest a pipeline that produces robust DE analysis results for sRNA transcripts, at least for categorical factors and two-group comparisons of biosamples.

Since each dataset was generated with different tissues, library preparation kits, and sequencing approaches (sequence machine and read length), large variations in data were observed, as expected. We aimed to use existing tools to construct an optimal pipeline for quality sequencing data analysis, despite the differences in input data.

To account for data variation in the original datasets, flexible trimming thresholds were applied. The input reads lengths were between 42 nt (“Donkin”) and 150 nt (“Hua”) across datasets. We suggest using 15 nt as a lower bound and Read length − 40% of adapter length as an upper bound of read length for trimming. This approach afforded good adapter removal and avoided significant loss of reads for a range of datasets.

There are many tools for the assignment of trimmed reads to obtain usable expression data. Assigning processes can be conducted in different ways, and each has its strength and limitations. Alignment-based methods seem to be less specialized, and we suggest using a bowtie aligner, which provides a high assignment rate and H-score for sRNAs across datasets.

We tested six thresholds for data filtering. As expected, thresholds based on the minimum counts of transcripts (5 and 10) resulted in low transcript numbers and may result in the loss of important data (Appendix A). Filtering by mean (counts) > 5 and median (counts) > 5 provides more transcripts, so these two thresholds were used for further analysis. The median threshold was stricter for skewed distributed transcripts and provided less significant findings after DE analysis (Appendix A). Gene signatures based on mean (counts) > 5 threshold had higher H-scores for alignment-based pipelines, so this cut-off is recommended for the analysis of two groups. Median (counts) > 5 threshold may be more useful for more complicated analyses, such as the analysis of continuous data with many covariates, but this was not formally tested here.

Three well-established packages based on regression analysis (DESeq2, edgeR, limma) and two groups of contrast (categorical) factors were used for DE analysis. To assess the quality of delivered expression signatures, the H-score metric of the Hobotnica package was employed. We observed a similar medium quality for the data separation test using three packages across datasets. We suggest using DESeq2 with multiple test corrections for data with strong and well-detected signals, and limma with no *p*-values adjustment for weaker-signal data.

Tools designed specifically for small RNA analysis (such as SPORTS or sRNAnalyzer) may seem to be more suitable for sRNA seq data analysis. A disadvantage of the sRNA-specific tools is the ’map and remove’ approach, where the order of databases used to sequentially align reads can affect the analysis outcome and different sRNA biotypes are not treated independently. The sRNAnalyzer cannot analyze tRNA or rRNA, and SPORTS cannot analyze piRNA, as depicted in the Figure 5. These limitations and outcomes make bioinformatic analysis less flexible and informative.

It should be noted that analyzed datasets demonstrated heterogeneity according to the processing steps. We suggest that the best pipeline depends on the input data (cell/tissue type, library preparation kit, sequencing approaches), hypothesis of the study, and sRNA-seq data (type of factors, whether categorical or continuous, how many covariates are used for adjustment). In this study, continuous factors and complex models with covariates were not analyzed. Nonetheless, for categorical factors and two groups of biosamples, our optimized pipeline for sRNA analysis recommends the following steps:Trimming with the lower length bound = 15 and the upper-length bound = Read length − 40% of Adapter length;Mapping on a reference genome with bowtie aligner, with one mismatch allowed (-v 1 parameter);Filtering by mean threshold > 5;DESeq2 for DE analysis with adjusted *p*-value < 0.05.

Although we investigated human miRNA, piRNA, tsRNA data, we anticipate that our optimized pipeline may be employed using similar sRNA data from other organisms.

## 4. Materials and Methods

### 4.1. Data Sources

Small RNA sequencing data (SRA archives) from 7 published human studies were retrieved from GEO database and extracted to FASTQ reads using sra-toolkit [65] (see Table 4). Raw reads’ quality was explored by FastQC [66]. Dataset from Wong et al., article [58] (referred to as “Wong”) consists of 120 fastq-files for 30 blood plasma samples. In this study, three small RNA library preparation kits (CleanTag, NEXTflex, QIAseq) and two RNA extraction methods (miRNeasy and MagnaZol) were compared. This dataset was used for small RNA seq data analysis benchmarking, but not for DE analysis.

A prospective case-control study from Huang et al., article [59] (referred to as “Huang”) was designed to identify the changes in expression of miRNA and mRNA, using 10 blood samples in dilated cardiomyopathy patients and 10 paired, healthy, control blood samples.

A total of 108 RNA sequencing samples from Delker et al., dataset [60] (referred to as “Delker”) from RNAlater preserved clinical biopsies (16 sessile serrated adenomas/polyps, 14 hyperlastic polyps, 14 adenomatous polyps, 34 uninvolved colon and 30 control colon samples) were used for small RNA seq data analysis benchmarking, and 2 contrasts (16 sessile serrated adenomoas/polyps vs. 14 hyperlastic polyps (”Delker-P“) and 15 right vs. 15 left control colon (”Delker-C“) samples) were used in DE analysis.

Six sperm samples were prospectively collected on a monthly basis from 17 healthy male participants for Morgan et al., study [61] (referred to as ”Morgan“). A total of 87 human sperm samples with high and low rates of good-quality embryos were analyzed in Hua et al., study [62] (referred to as “Hua“). The semen of 13 lean and 10 obese individuals were analyzed in Donkin et al.’s study [63] (referred to as “Donkin“). Pure fractions of motile spermatozoa collected from 12 young healthy individuals before and after 6 weeks of endurance training, and after 3 months without exercise, were analyzed in Ingerslev et al. [64] study (referred to as ”Ingerslev“).

### 4.2. Preprocessing of RNA-seq Data

All read adapters were removed with cutadapt [37], following lab protocols [67,68,69]. Reads of less 15 nt in length (lower bound) were removed, since a smaller read length makes aligning and expression quantification difficult and not robust. To adjust the pipeline and derive optimal parameters, we used several trimming options for the upper bound. Reads without the upper read length bound and with differently varied thresholds were processed as follows (see Figure 2A):45 ntRead length − X × Adapter lengthwhere X = 10%, 20%, 30%, ..., 100%Read length − X ntwhere X = 3, 6, 9, ..., 30 ntRead length × (1 − X $\frac{\mathrm{Read}\;\mathrm{length}}{\mathrm{Adapter}\;\mathrm{length}})$where X = 0.05, 0.1, 0.15, ..., 0.5

Trimmed read length distributions were analyzed to infer the optimal threshold that can preserve the best sequencing data signal. One of the trimming strategies with a suitable performance was chosen (see Section 2).

### 4.3. Processing of Small RNA-seq Data

#### 4.3.1. sRNAs

The trimmed reads were processed with various alignment and pseudoalignment methods. Alignment methods include mapping on hg38 [70] reference genome with bowtie [40] (with 0 and 1 mismatch), hisat2 [38] and STAR [39] aligners.

Only one best alignment was determined using the bowtie aligner for every read that was to be used in further analysis: bowtie -x genome_index - q in.fq -S out.sam -v {0,1} -m 100 -k 1 –best –strata

The Hisat2 aligner was used with its standard parameters but without spliced alignments and softclip: hisat2 -x genome_index -U in.fq -S out.sam –no-spliced-alignment –no-softclip STAR aligner was also used with its standard parameters without allowing introns and mismatches: STAR –runMode alignReads –genomeDir genome_index –readFilesIn in.fq –outFileNamePrefix out –outFilterMismatchNmax 0 –alignIntronMax 0 –alignIntronMin 0 –genomeLoad LoadAndKeep.

For transcript abundance quantification, featureCounts [42] from Rsubread was used. At this stage, Integrated Transcript Annotation of Small RNA (ITAS) [30], which contains filtered transcripts of different sRNA biotypes, including miRNA, tRNA, tsRNA, piRNA and rRNA, was employed.

Other methods, including transcriptome aligner RSEM [44], and pseudoaligners kallisto [45] and salmon [46] with different kmer lengths, were also applied, along with their default parameters, to align data reads with ITAS transcripts.

#### 4.3.2. tRNA Fragments

As several variants of the same fragment need to be considered, the quantification and assigning of reads to tRNA fragments is a probabilistic procedure. Therefore, a probabilistic aligner, such as kallisto, is a better choice for this part of the analysis.

Trimmed reads were mapped on the hg38 reference genome with the various described approaches and those transcripts that were assigned to tRNA by Rsubread were extracted and processed using several k-mer values kallisto [45] for probabilistic mapping on ITAS tsRNA fragments sequences.

#### 4.3.3. sRNA Tools

All samples, once trimmed and pre-processed as described above, were also processed using sRNA analysis tools SPORTS [26] and sRNAnalyzer [25] with the default parameters. All preprocessing steps, including quality and size filtering, and filtering for contaminates, were conducted using these tools.

SPORTS1.0 tool was used with default databases: reads from all analyzed human samples were sequentially ”mapped and removed“ to the sRNA databases (miRBase, rRNA database (collected from NCBI), GtRNAdb, piRNA database, Ensembl and Rfam). The only parameter used in command was -i for the path to the input file; -g, -m, -r, -t, -w for paths to genome, miRNA, rRNA, tRNA and piRNA files; -p for the number of involved processors. A summary text file was used as the list of gene counts for each sample.

The sRNAnalyzer was launched with implemented databases “small RNA Databases” and “Human and Exogenous Databases”. “NCBI Non-Human Databases” was not used, as it was redundant, and all samples were pre-filtered before starting sRNAnalyzer. The only modified file was DB_cofig.conf. Pipeline_config.yaml file was not modified. As output, “XX.feature” file with summary counts for each sample was used after analysis.

### 4.4. Filtering Thresholds

Transcript filtering based on expression values is an important step in DE analysis, as sRNA transcripts with a low coverage and low expression may add noise to the signal, thus decreasing the robustness of the analysis.

Three filtering strategies were applied:Min filtering: expression value for a transcript in all samples is higher than the threshold (N)min(counts) > (N);Mean filtering: mean expression value for a transcript in all samples is higher than the threshold (N)mean(counts) > N;Median filtering: median expression value for a transcript in all samples is higher than the threshold (N)median(counts) > N.

Two threshold values, N = 5 and N = 10, were applied.

### 4.5. Differential Expression

Transcripts from the counts matrix were filtered using several thresholds: min count > {5,10}, mean count > {5,10}, median count > {5,10}. For DE analysis, data filtered by mean and median count > 5 were used. DE analysis was conducted using DESeq2 [47], edgeR [48] and limma [49] packages. Datasets (from ”Huang“, ”Hua“, ”Donkin“ papers) or subdatasets from ”Delker“ and ”Ingerslev“ papers consisting of two groups were used. The following types of threshold were obtained:*p*-value < 0.05;Adjusted *p*-value < 0.05.

DE results in an expression signature, i.e., a list of transcripts with significant differences in expression between considered conditions. The following options are usually used for thresholds in the processing of the initial signature:Adjusted *p*-value threshold (preferable);Combination of *p*-value and absolute fold change thresholds.

Multiple testing correction is usually used in DE analysis. However, due to the low signal in sRNA expression data, many studies (e.g., ”Morgan“, ”Huang“) only use the unadjusted *p*-value. Although this implies insufficient statistical power in the analysis, the results may suggest candidate transcripts for further exploration.

### 4.6. Expression Signature Quality Evaluation

The number of DE transcripts cannot be used as a robust metric for DE analysis quality, as this approach does not account for false-positive results. To evaluate the expression signature quality, we applied the Hobotnica [71] approach.

In this process, the candidate expression signatures delivered after the application of differing preprocessing and processing procedures, such as a differential model or genome aligner, were inferred to the expression data to produce a distance matrix between samples. Next, the distance matrix was compared with the expected samples’ relationship structure, given the known groups of samples. The results of expression signature quality evaluation, presented as H-score value, reveal of the quality of a candidate signature’s data separation. H-score can be scaled from 0 (the worst) to 1 (the best).

### 4.7. Stages and Metrics of Benchmarking

We used metrics for each stage of RNA-seq data analysis, from the biosampling and library preparation to the DE. These are described in Table 5. The lengths of input reads and adapters were used to estimate the upper bound of trimming. All reads from the analyzed datasets had suitable quality, as assessed by the reads’ quality threshold, an important metric for sRNA data. For the alignment-based pipelines, the alignment rate was calculated as the ratio of aligned reads and input reads, and the assignment rate was calculated as the ratio of assigned reads and aligned reads. The assignment rate for sRNA-based and pseudoalignment-based pipelines was calculated as the ratio of assigned reads and input reads. After conducting a suitable pipeline, all expected sRNA biotypes should be returned. The filtering stage is important for the removal of non-expressed transcripts, but should retain sufficient transcript numbers for subsequent DE analysis. The DE metric is the number of significant transcripts, while the H-score characterizes the expression signature quality.

Different post-DE downstream applications may be applied to the retrieved expression signatures. These include the Connectivity Map [72,73], GO terms [74] and KEGG [75] pathways analysis, hallmark signatures inference [76], and genome browser analysis [77]. Although these approaches provide useful and valuable information, they cannot be used for benchmark purposes, as appropriate metrics for the quality of analysis cannot be robustly introduced [57].

## 5. Conclusions

In this study, the effect of various factors that impact the expression analysis of human sRNA at different stages of data processing were investigated. The optimal pipeline alternatives and their parameters were identified, and an optimized pipeline for setting and running sRNA expression analysis was proposed. Assessing the resulting expression signatures with rank-statistics-based inference suggests a way to estimate the quality of resulting signatures and performance of bioinformatical analysis for particular biological data.

## Figures and Tables

**Figure 1 ijms-24-04195-f001:**
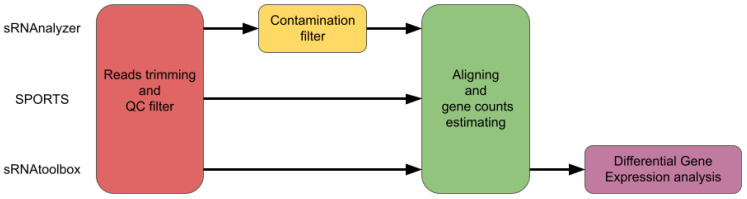
Pipeline comparison for sRNAnalyzer, SPORTS and sRNAtoolbox.

**Figure 5 ijms-24-04195-f005:**
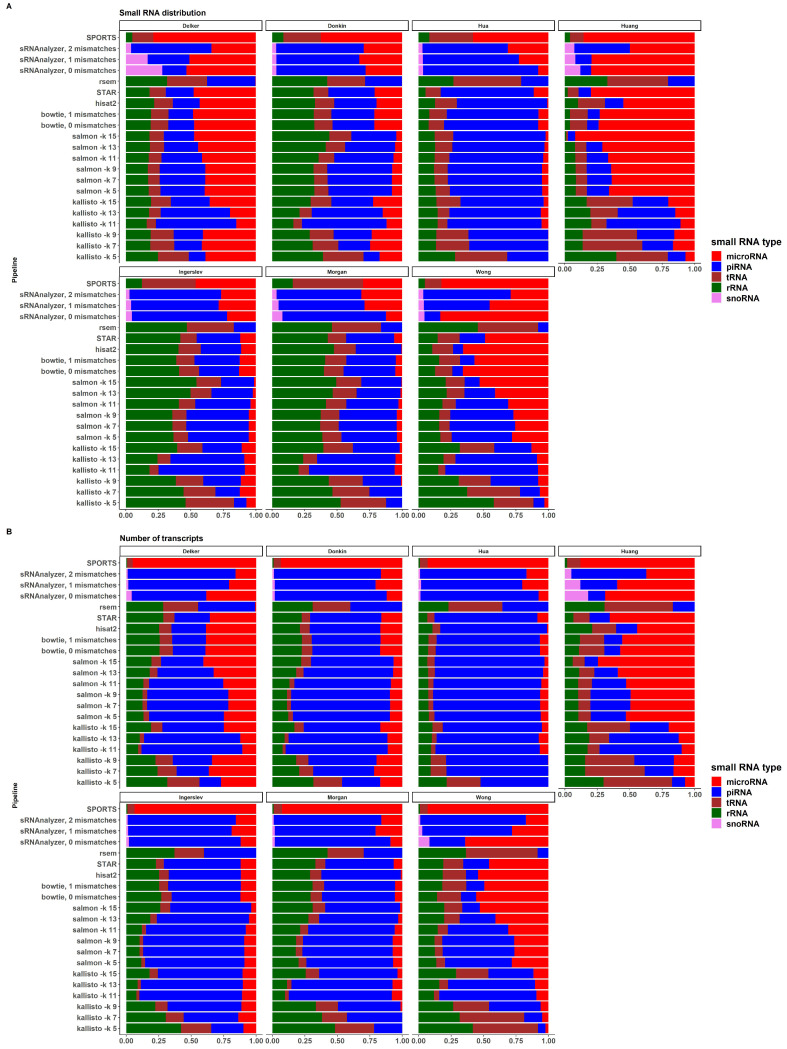
(**A**) Distribution of sRNA types’ expression values by datasets [58,59,60,61,62,63,64] and pipelines (**B**) Distribution of sRNA types’ transcripts by datasets [58,59,60,61,62,63,64] and pipelines.

**Figure 6 ijms-24-04195-f006:**
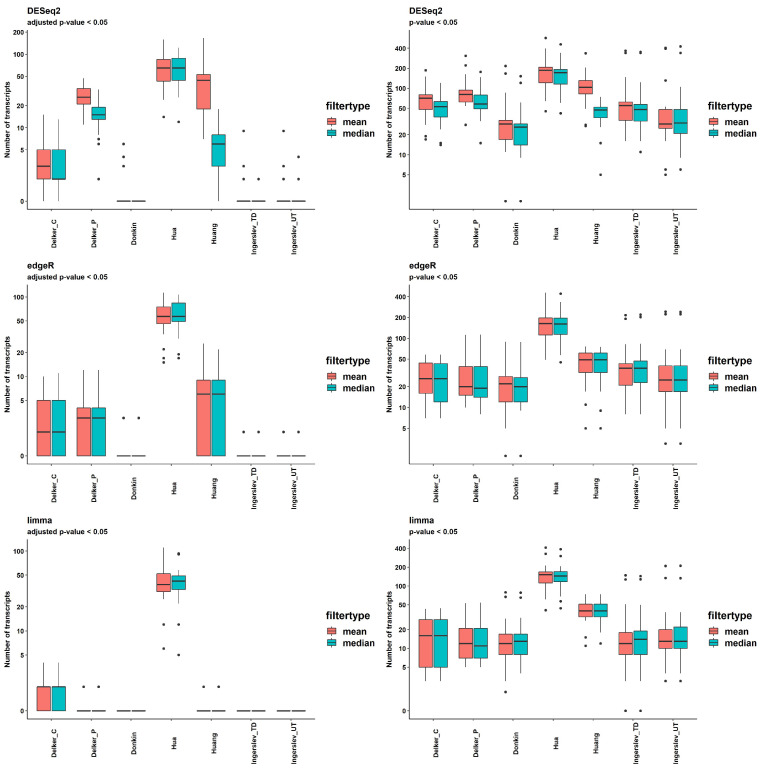
Number of DE transcripts provided by DESeq2, edgeR and limma with *p*-value or adjusted *p*-value < 0.05 among data [59,60,62,63,64] with different filtering.

**Table 1 ijms-24-04195-t001:** Main features of some existing sRNAs pipelines.

Name	Year	Last Update	Status	Interface	Type	RNA Classes	Output
DSAP	2010	2010	not supported	GUI	map and	ncRNA (Rfam), miRNA	DE
			for Solexa only		remove		transcripts
Oasis 2	2018	2018	not supported	online	map and	miRNA, piRNA,	transcripts
					remove	small nucleolar RNA (snoRNA)	counts
iSmaRT	2017	2017	supported	GUI	map and	miRNA, piRNA	DE
			from ftp-server		remove		transcripts
sRNAPipe	2018	2021	supported	GUI	map and	miRNA, piRNA,	transcripts
			Galaxy-based		remove	rRNA, tRNAs,	counts
						transposable elements	
iSRAP	2015	2016	supported	CLI	map and	miRNA, piRNA, snoRNA	DE
					remove		transcripts
miARma-seq	2016	2019	not supported	CLI	map and	miRNA, snoRNA	DE
					remove		transcripts
tsRFun	2022	2022	supported	online	map and	tsRNA	DE
					remove		transcripts
piPipes	2015	2016	supported	CLI	map	piRNA	transcripts
							counts
PILFER	2018	2018	supported	CLI	map	piRNA	transcripts
							counts
miRanalyzer	2011	2014	not supported	online	map and	miRNA	DE
					remove		transcripts
sRNA workbench	2018	2018	not supported	CLI	map	miRNA	alignment
sRNAtoolbox	2022	2022	supported	online	map and	miRNA, tRNA,	DE
					remove	ncRNA, cDNA	transcripts
sRNAnalyzer	2017	2017	supported	CLI	map and	miRNA, piRNA, tRNA,	transcripts
					remove	snoRNA	counts
SPORTS	2018	2021	supported	CLI	map and	miRNA, piRNA, rRNAs,	transcripts
					remove	tRNAs, tRNA fragments	counts

**Table 2 ijms-24-04195-t002:** Recommended pipeline for sRNA analyses.

Stage	Pipeline Command	Justification
Trimming	Read length: lower bound-15 and	Retain sufficient and the same number of reads after
	upper bound-Read length − 40% of adapter length	trimming for downstream analyses for all datasets.
Aligning	bowtie aligner with 1 mismatch allowed	The high alignment rate and H-score for all datasets.
Assigning	ITAS [30]	Optimized annotation for small RNA.
Filtering	mean count > 5	Sufficient number of transcripts for the
		downstream analysis and higher H-score.
DE analysis	DESeq2	Sufficient number of significant transcripts and
		high H-score.

**Table 3 ijms-24-04195-t003:** Results for bowtie -v 1 pipeline; *—assigned/aligned; **—mean (counts) >5; ***—DESeq2 [58,59,60,61,62,63,64].

Dataset	Biological Object and Contrast	Alignment Rate	Assignment * Rate	Number of Filtered Transcripts **	Number of Findings ***	H-Score
”Hua“	Sperm	0.94	0.29	901	71	0.66
”Donkin“	Sperm	0.97	0.07	966	0	-
”Ingerslev“	Sperm untrained contrast	0.9	0.06	1236	0	-
	detrained contrast				0	-
“Morgan”	Sperm	0.89	0.1	-	-	-
“Delker”	colon cancer polyps contrast	0.94	0.55	1142	21	0.6
	controls contrast				5	0.86
“Huang”	Blood	0.94	0.48	498	17	0.73
“Wong”	Blood plasma	0.68	0.38	-	-	-

**Table 4 ijms-24-04195-t004:** Description of the publicly available human datasets that were used in this study for benchmarking.

GEO ID	Cite	Object	Total Samples Number (Contrast Groups)	Raw Reads Length	Library Kit
GSE118125					
https://www.ncbi.nlm.nih.gov/geo/query/acc.cgi?acc=GSE118125					
(accessed on 11 August 2022)	[58]	Blood plasma	30	76	NEXTflex
					CleanTag, Qiaseq
GSE117841					
https://www.ncbi.nlm.nih.gov/geo/query/acc.cgi?acc=GSE117841					
(accessed on 11 August 2022)					
	[59]	Blood	20 (10 vs. 10)	50	Truseq
GSE118504					
https://www.ncbi.nlm.nih.gov/geo/query/acc.cgi?acc=GSE118504					
(accessed on 11 August 2022)					
	[60]	Colon Cancer	108 (16 vs. 14 and 15 vs. 15)	50	NEBNext, Truseq
GSE159155					
https://www.ncbi.nlm.nih.gov/geo/query/acc.cgi?acc=GSE159155					
(accessed on 11 August 2022)					
	[61]	Sperm	98	50	Truseq
GSE110190					
https://www.ncbi.nlm.nih.gov/geo/query/acc.cgi?acc=GSE110190					
(accessed on 11 August 2022)					
	[62]	Sperm	87 (64 vs. 23)	150	Illumina
GSE74426					
https://www.ncbi.nlm.nih.gov/geo/query/acc.cgi?acc=GSE74426					
(accessed on 11 August 2022)					
	[63]	Sperm	23 (13 vs. 10)	42	NEBNext
GSE109478					
https://www.ncbi.nlm.nih.gov/geo/query/acc.cgi?acc=GSE109478					
(accessed on 11 August 2022)					
	[64]	Sperm	24 (9 vs. 9 and 9 vs. 6)	51	NEBNext

**Table 5 ijms-24-04195-t005:** Metrics for benchmarking according to stage of bioinformatic analysis.

Stage	Metrics
Input data	Biological object, lab kit, read length, adapter length, reads quality
Trimming	Part of reads-processed trimming, length threshold
Aligning (for genome-alignment-based pipelines)	Alignment rate
Assigning	Assignment rate, distribution by sRNA type
Filtering	Number of transcripts after trimming
DE	Number of significant transcripts
Expression signature quality evaluation	H-score

## Data Availability

Publicly available datasets were analyzed in this study. These data can be found here: https://www.ncbi.nlm.nih.gov//geo/query/acc.cgi?acc=GSE110190 GSE110190 (accessed on 11 August 2022), https://www.ncbi.nlm.nih.gov/geo/query/acc.cgi?acc=GSE74426 GSE74426 (accessed on 11 August 2022), https://www.ncbi.nlm.nih.gov/geo/query/acc.cgi?acc=GSE109475 GSE109475 (accessed on 11 August 2022), https://www.ncbi.nlm.nih.gov/geo/query/acc.cgi?acc=GSE159155 GSE159155 (accessed on 11 August 2022), https://www.ncbi.nlm.nih.gov/geo/query/acc.cgi?acc=GSE118125 GSE118125 (accessed on 11 August 2022), https://www.ncbi.nlm.nih.gov/geo/query/acc.cgi?acc=GSE117841 GSE117841 (accessed on 11 August 2022), https://www.ncbi.nlm.nih.gov/geo/query/acc.cgi?acc=GSE118504 GSE118504 (accessed on 11 August 2022).

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
