# Peer review of "Approaches for sRNA Analysis of Human RNA-Seq Data: Comparison, Benchmarking"

_ijms, 2023, doi:10.3390/ijms24044195_

Round 1

Reviewer 1 Report

Bioinformatic tools that are designed for small RNA analysis like sRNAnalyzer or SPORTS have a major disadvantage in the form that the the order of databases used to sequentially align reads can affect the analysis outcome, and different sRNA biotypes are not treated independently.

In order to address this issue, the authors have come up with a pipeline that can be used to solve this problem. In this review, the authors have discussed about the advantages and disadvantages of using bioinformatic approaches/ software that are used to map the sRNA and measure their differential expression. In order to prove their point, the authors have used three previously published dataset and subjected them to bioinformatic analysis using different tools. There are two essential components of the small RNA analysis: mapping the reads to the genome and measuring their differential expression. To map the reads, the first step involved checking the quality of the reads using FASTQC, followed by trimming of the reads that show poor quality scores at both ends using bioinformatic tools like trimomatic as well as cutadapt. This step is followed by removing the adapters that used during generation of RNA library preparation. Finally, the reads were mapped to the reference genome using the aligner bowtie 2 where only one mismatch was allowed as a cut-off to mapping the reads. This step is followed by data filtering. In order to do a comparative analysis, the authors have used 6 different methods and based on their observation, they suggested that filtering by mean gave them the best coverage. Finally, the authors measured the differential expression with DESeq2.  

Overall, this review will be able to help thousands of researchers analyze their sRNA analysis in the correct way and should be of great help.

However, I’ve couple of suggestions that the authors should incorporate in their analysis:

1.     Can this method be used in bacterial sRNA analysis? The authors should incorporate a section on this as this will make the review more inclusive. It would great if the authors can incorporate a bacterial dataset and do a similar analysis and show how their pipeline can benefit bacterial sRNA analysis.

2.     The sRNA can be also visualized using the genome browser and can be interpret the mechanism of gene regulation. The authors should incorporate how this analysis can be used in this regard.

3.     Also, did the authors see any differences when using different aligners to map the reads. The authors should incorporate couple of lines regarding this.

Author Response

Open Review 1

 Response:

We thank the reviewers for their thoughtful and constructive comments on our paper.  We offer a point-by-point response to each comment below. The revised text of the manuscript was highlighted in red.

We have also followed the Reviewers’ request to improve English. Looking forward to your consideration.

Oleg Sergeyev, on behalf of co-authors.

English language and style

( ) English very difficult to understand/incomprehensible
( ) Extensive editing of English language and style required
( ) Moderate English changes required
(x) English language and style are fine/minor spell check required
( ) I don't feel qualified to judge about the English language and style

Yes

Can be improved

Must be improved

Not applicable

Does the introduction provide sufficient background and include all relevant references?

(x)

( )

( )

( )

Are all the cited references relevant to the research?

(x)

( )

( )

( )

Is the research design appropriate?

( )

(x)

( )

( )

Are the methods adequately described?

( )

(x)

( )

( )

Are the results clearly presented?

(x)

( )

( )

( )

Are the conclusions supported by the results?

(x)

( )

( )

( )

Comments and Suggestions for Authors

Bioinformatic tools that are designed for small RNA analysis like sRNAnalyzer or SPORTS have a major disadvantage in the form that the the order of databases used to sequentially align reads can affect the analysis outcome, and different sRNA biotypes are not treated independently.

In order to address this issue, the authors have come up with a pipeline that can be used to solve this problem. In this review, the authors have discussed about the advantages and disadvantages of using bioinformatic approaches/ software that are used to map the sRNA and measure their differential expression. In order to prove their point, the authors have used three previously published dataset and subjected them to bioinformatic analysis using different tools. There are two essential components of the small RNA analysis: mapping the reads to the genome and measuring their differential expression. To map the reads, the first step involved checking the quality of the reads using FASTQC, followed by trimming of the reads that show poor quality scores at both ends using bioinformatic tools like trimomatic as well as cutadapt. This step is followed by removing the adapters that used during generation of RNA library preparation. Finally, the reads were mapped to the reference genome using the aligner bowtie 2 where only one mismatch was allowed as a cut-off to mapping the reads. This step is followed by data filtering. In order to do a comparative analysis, the authors have used 6 different methods and based on their observation, they suggested that filtering by mean gave them the best coverage. Finally, the authors measured the differential expression with DESeq2.  

Overall, this review will be able to help thousands of researchers analyze their sRNA analysis in the correct way and should be of great help.

However, I’ve couple of suggestions that the authors should incorporate in their analysis:

  1. Can this method be used in bacterial sRNA analysis? The authors should incorporate a section on this as this will make the review more inclusive. It would great if the authors can incorporate a bacterial dataset and do a similar analysis and show how their pipeline can benefit bacterial sRNA analysis.

Response :

Although the goal of the paper was to identify the optimal pipeline configurations for each step of sRNA analysis for human data (we clarified it for Readers in the Title, line 7 of Abstract and further), we agree with the Reviewer that a similar goal would be important for other species including bacteria. At the same time aiming to answer the Reviewer’s question, we have conducted exploratory search of small RNA bacterial data. We used key words “Bacterial”, “Small RNA”, “Database” in Pubmed. We have found the following recent papers that we can categorize in two groups. First, papers based on the analysis of small RNA in bacteria and second, papers based on the search for bacterial small RNA among small RNA data of human or other animals.

  1. Analysis of small RNA in bacteria:
  • PresRAT: a server for identification of bacterial small-RNA sequences and their targets with probable binding region (PMC8244778)
  • Sequence-based bacterial small RNAs prediction using ensemble learning strategies (PMC6302447)
  • Bioinformatics analysis of small RNAs in Helicobacter pylori and the role of NAT-67 under tinidazole treatment (PMC7339756)
  • A cohabiting bacterium alters the spectrum of short RNAs secreted by Escherichia coli (PMID30376063)
  • Comparative global gene expression analysis of biofilm forms of Salmonella Typhimurium ATCC 14028 and its seqA mutant (PMID36470486)
  • NsiR3, a nitrogen stress‐inducible small RNA, regulates proline oxidase expression in the cyanobacterium Nostoc sp. PCC 7120 (PMID32799414)
  • Global discovery of small RNAs in the fish pathogen Edwardsiella piscicida: key regulator of adversity and pathogenicity (PMC6288947)
  • Isolation and Characterization of a microRNA-size Secretable Small RNA in Streptococcus sanguinis (PMID27796789)
  1. Search for bacterial small RNA among small RNA data of human or other animals
  • BIC: a database for the transcriptional landscape of bacteria in cancer (PMC9825443)
  • SEAweb: the small RNA Expression Atlas web application (PMC6943056)
  • Characterisation of the Small RNAs in the Biomedically Important Green-Bottle Blowfly Lucilia sericata (PMC4372549)
  • Quantitative mapping of the cellular small RNA landscape with AQRNA-seq (PMC8355021)

However, none of the above mentioned publications contain data that meets the criteria to be processed by the pipelines of our study:

  • presence of untrimmed fastq in public access,
  • targeted sequence data of various types of small RNA (smaller 50-100 nt; from 15 nt),
  • presence of at least two groups of samples for analysis of differential expression.

Pipelines for human sRNA data include mapping data for several types of sRNA. However, some of them, piRNA and tRNA-derived small RNA, are poorly investigated for bacteria.

Thus, the Reviewer's suggestion to add bacterial datasets to our benchmark study cannot be carried out, unfortunately, at least now. We have added clarification in the title, abstract, introduction, result and conclusion that we used human data. At the same time we have added in Discussion, lines 512-513 following text: “Although we investigated human microRNA, piRNA, tsRNA data we anticipate that our optimized pipeline be employed with similar sRNA data from other organisms”.

  1. The sRNA can be also visualized using the genome browser and can be interpret the mechanism of gene regulation. The authors should incorporate how this analysis can be used in this regard. -

Response: We agree with the Reviewer that the interpretation and downstream analysis of differential expression using genome browser and enrichment analysis, is certainly important. Our goal in this study, however, was to perform the benchmark of the analysis methods and compare those based on particular, clearly interpretable quantitative metrics (please see Table 3). Downstream applications, among which is the one the Reviewer suggested, cannot provide such metrics for the approaches comparison. For this reason, we deliberately chose to limit our pipeline to identification of differentially expressed genes.

We described it in detail in the Methods, lines 273-278: “Different post differential expression downstream applications may be applied to the retrieved expression signatures. Among them are Connectivity Map (Lamb, 2007; Musa et al, 2018), GO terms (Young et al, 2012) and KEGG (Kanehisa et al, 2007) pathways analysis, hallmark signatures inference (Liberzon et al, 2011), and genome browser analysis (Karolchik et al, 2003). Although these approaches provide useful and valuable information, they cannot be used for the benchmark purposes, as appropriate metrics for the analysis quality cannot be robustly introduced (Stupnikov et al, 2021).

  1. Also, did the authors see any differences when using different aligners to map the reads. The authors should incorporate couple of lines regarding this.

Response: We agree that comparative results of using various aligners are important. We presented it for the genome-alignment-based methods (bowtie, hisat2 and STAR) and prebuilt transcriptome alignment-based method (RSEM) in the section entitled “Genome or transcriptome aligning”, lines 310-348, and in the Figures 3,4, Suppl Tables 2,3 and Suppl Figures 3,4.

Submission Date

07 December 2022

Date of this review

16 Dec 2022 22:13:12

Reviewer 2 Report

In addition to the suggestions by the reviews, the article needs to be edited throughout for spelling, punctuation and grammar. Also please verify that citations are used appropriately throughout.

Author Response

Open Review 2

English language and style

( ) English very difficult to understand/incomprehensible
( ) Extensive editing of English language and style required
( ) Moderate English changes required
(x) English language and style are fine/minor spell check required
( ) I don't feel qualified to judge about the English language and style

Yes

Can be improved

Must be improved

Not applicable

Does the introduction provide sufficient background and include all relevant references?

(x)

( )

( )

( )

Are all the cited references relevant to the research?

(x)

( )

( )

( )

Is the research design appropriate?

(x)

( )

( )

( )

Are the methods adequately described?

(x)

( )

( )

( )

Are the results clearly presented?

( )

(x)

( )

( )

Are the conclusions supported by the results?

( )

(x)

( )

( )

Comments and Suggestions for Authors

In addition to the suggestions by the reviews, the article needs to be edited throughout for spelling, punctuation and grammar. Also please verify that citations are used appropriately throughout.

Response:

Per Revewer's suggestions, we have revised the Results and Conclusion section. Additionally, we once again carefully checked the article with the help of a co-author who is a native English speaker and verified that citations are used appropriately.

Submission Date

07 December 2022

Date of this review

07 Jan 2023 12:43:35

Reviewer 3 Report

Bezuglov et al. aimed to identify the optimal pipeline configurations for each step of sRNA analysis, including reads trimming, filtering, mapping, transcript abundance quantification and differential expression analysis. This study is interesting, they suggested a way to estimate the quality of resulting signatures and performance of bioinformatical analysis for a particular biological data. 

1. For data involved in this study, it is better to contain sequencing data with larger sample sizes. Why not select sequencing data from TCGA? Larger sample sizes and sequencing data with high-quality may be an important factor to estimate the different algorithms.

2. For the screening differentially expression analysis, the current data contained fewer sample sizes. It is difficult to assess DE genes based on these limited data. Furthermore, it is better if the selected data contain more sequencing data from tissues, which are main data sources for the relevant study. 

3. In Table 2, the detailed tumor or control sample sizes should be provided. 

4. It is better to show more analysis or results of the DE analysis. The DE analysis may be a critical step for most studies, which can provide more references for further analysis and experimental validation.

5. It is better to show the detailed flowchart for this study based on the current Figure 1. Also, authors selected several algorithms to perform analysis, why not select more algorithms to perform analysis? For example, several famous algorithms, including DEseq, edge, limma, and so on, and these algorithms were only mentioned in the discussion section without corresponding results.

Author Response

Open Review 3

English language and style

( ) English very difficult to understand/incomprehensible
( ) Extensive editing of English language and style required
( ) Moderate English changes required
(x) English language and style are fine/minor spell check required
( ) I don't feel qualified to judge about the English language and style

Yes

Can be improved

Must be improved

Not applicable

Does the introduction provide sufficient background and include all relevant references?

( )

(x)

( )

( )

Are all the cited references relevant to the research?

(x)

( )

( )

( )

Is the research design appropriate?

(x)

( )

( )

( )

Are the methods adequately described?

( )

(x)

( )

( )

Are the results clearly presented?

( )

(x)

( )

( )

Are the conclusions supported by the results?

( )

(x)

( )

( )

Comments and Suggestions for Authors

Bezuglov et al. aimed to identify the optimal pipeline configurations for each step of sRNA analysis, including reads trimming, filtering, mapping, transcript abundance quantification and differential expression analysis. This study is interesting, they suggested a way to estimate the quality of resulting signatures and performance of bioinformatical analysis for a particular biological data. 

  1. For data involved in this study, it is better to contain sequencing data with larger sample sizes. Why not select sequencing data from TCGA? Larger sample sizes and sequencing data with high-quality may be an important factor to estimate the different algorithms.

Response: We agree with the Reviewer that large sample sizes and sequencing data with high-quality are important factors to estimate the different algorithms. The largest sRNA dataset used in our study is “Hua” dataset, which contains sRNA data for 64 vs 23 human sperm samples (Table 2 and lines 146-148).

The Cancer Genome Atlas (TCGA), as the Reviewer points out, collected many types of data for  over 20,000 tumor and normal samples. However, it contains small RNA data only for miRNA sequencing (not for all small non-coding RNA) and is using BAM-format (not in fastq-format), https://www.cancer.gov/about-nci/organization/ccg/research/structural-genomics/tcga/using-tcga/types). Since we sought to benchmark the data derived from raw untrimmed sequencing data, TCGA data, unfortunately, was not suitable for the analysis in our paper.

Publicly available small RNA seq datasets with untrimmed fastq-files suitable for differential expression analysis are limited. We conducted another search, using GEO Datasets and recent literature and have found larger dataset of 822 fastq files. In that set there are 126 samples from "normal" patients and 114 from cancer patients (https://www.ncbi.nlm.nih.gov/geo/query/acc.cgi?acc=GSE110381, PMID29490987). However, we were not able to identify how to stratify fastq files by groups using that dataset.

  1. For the screening differentially expression analysis, the current data contained fewer sample sizes. It is difficult to assess DE genes based on these limited data. Furthermore, it is better if the selected data contain more sequencing data from tissues, which are main data sources for the relevant study.

Response: The Reviewer raises an important point of the sample size. Indeed, it was demonstrated, the sample size of analyzed data can affect the results of differential gene expression drastically. Some of these studies were conducted earlier by some of the coauthors of this paper (Stupnikov et al, 2021). The largest sRNA dataset that we used in the study is “Hua” dataset which contains sRNA data for 64 vs 23 human sperm samples (Table 2 and lines 146-148). Per Reviewer suggestion we have analyzed literature and GEO datasets and have found a larger dataset of 822 fastq files, in which there are 126 samples from "normal" patients and 114 from cancer patients (https://www.ncbi.nlm.nih.gov/geo/query/acc.cgi?acc=GSE110381, PMID29490987). However we were not able to identify how to stratify fastq files by groups using that dataset. Given this fact, the Reviewer's suggestion to add datasets with larger sample count to our benchmark study cannot be carried out at that step unfortunately.

  1. In Table 2, the detailed tumor or control sample sizes should be provided.

Response: Thank you for pointing it. We have clarified the column “Total samples number (contrast groups)” and presented the number of samples in compared groups for studies (Table 2).

  1. It is better to show more analysis or results of the DE analysis. The DE analysis may be a critical step for most studies, which can provide more references for further analysis and experimental validation.

Response: We agree that comparative results of using various packages for differential expression analysis are important. We presented it for packages DESeq2, edgeR and limma using two options of filtering and two thresholds of significance in the section entitled “Differential expression”, lines 374-398, and in the Supplemental Figures 7 and Supplemental Table 5. To be clearer for readers we moved Supplemental Figure 8 in the main body of the manuscript (now it is Figure 6) presenting number of significant differentially expressed transcripts using three DE packages (DESeq2 - top; edgeR - middle; limma - bottom)  and two thresholds of significance (adjusted p-value <0.05 - left; p-value<0.05 - right) by two filtering (mean>5 and median>5) and by contrast.

  1. It is better to show the detailed flowchart for this study based on the current Figure 1. Also, authors selected several algorithms to perform analysis, why not select more algorithms to perform analysis? For example, several famous algorithms, including DEseq, edge, limma, and so on, and these algorithms were only mentioned in the discussion section without corresponding results.

Response: Per Reviewer’s suggestion we revised Figure 1 to be clearer to Readers.

Regarding famous algorithms for differential expression, including DESeq2, edgeR and limma, please see our response to the comment #4 above, We presented results for packages DESeq2, edgeR and limma using two options of filtering and two thresholds of significance in the section entitled “Differential expression”, lines 374-398, and in the Supplemental Figures 7 and Supplemental Table 5 and moved Supplemental Figure 8 in the main body of manuscript (now is Figure 6).

Thus, per Revewer's suggestions, we have revised the Introduction, Methods, Results, Discussion and Conclusion sections. Additionally, we once again carefully checked the manuscript with the help of a co-author who is a native English speaker and verified that citations are used appropriately

Submission Date

07 December 2022

Date of this review

20 Jan 2023 04:33:35

Round 2

Reviewer 1 Report

The authors have answered all the queries and the manuscript now looks much better.

Reviewer 3 Report

The authors have addressed all my questions.